

# A Better Understanding of an Extremely High Ozone Episode with Ensemble Simulation

Jinhui Gao[1, 2], Hui Xiao[3, 4]

[1]Chengdu Plain Urban Meteorology and Environment Observation and Research Station of Sichuan Province, School of
Atmospheric Science, Chengdu University of Information Technology, Chengdu 610225, China
[2]China Meteorological Administration Aerosol-Cloud and Precipitation Key Laboratory, Nanjing University of Information
Science & Technology, Nanjing 210044, China
[3]Guangzhou Institute of Tropical and Marine Meteorology, China Meteorological Administration, Guangzhou 510641,
China
[4]Guangdong Early Warning Centre (Guangdong Weather Modification Centre), Guangzhou 510641, China

*Correspondence to*: Hui Xiao (xh_8646@163.com)

**Abstract.** Severe ozone pollutions may occur in the Great Bay Area (GBA) when typhoons approach South China. However, numerical models often fail to capture the high ozone concentrations during the episodes, leading to uncertainties in

understanding their formation mechanisms. This study conducted an ensemble simulation with 30 members (EMs) using the WRF-Chem model, coupled with a self-developed ozone source apportionment method, to analyze an extremely high ozone episode associated with Typhoon NIDA in the summer of 2016. The newly proposed index effectively distinguished between well-performing (good) and poorly performing (bad) EMs. Compared to the bad EMs, the good EMs accurately reproduced surface ozone variations, particularly capturing the extremely high concentrations observed in the afternoon of

July 31. The formation of such high ozone levels was attributed to the retention of ozone in the residual layer at night and the enhanced photochemistry during daytime. As Typhoon NIDA approached, weak winds confined large amounts of ozone in the residual layer at night. The development of planetary boundary layer (PBL) facilitated the downward transport of ozone aloft, contributing to the rapid increase in surface ozone in the following morning. The enhanced photochemistry was primarily driven by increased ozone precursors resulting from favorable accumulation conditions and enhanced biogenic

emissions. During the period of high ozone concentrations, contributions from local and surrounding regions increased. Additionally, ozone from southeastern Asia could transport to the GBA at high altitudes and then contribute to surface ozone when the PBL developed.

## 1 Introduction

Tropospheric ozone, especially within the planetary boundary layer (PBL), has attracted much public attention because of its detrimental effects on human health and vegetation (Monks et al., 2015; Feng et al., 2019; Lu et al., 2020). As a typical



secondary air pollutant, tropospheric ozone is primarily formed via photochemical reactions with the participation of nitrogen oxides (NOx) and volatile organic compounds (VOCs) (Sillman, 1995; Xie et al., 2014; Li et al., 2019; Wang et al., 2019). Furthermore, the meteorological conditions play important roles in the chemical production and accumulation of

ozone in the atmosphere, leading to severe air pollution (Ding et al., 2013a; Wang et al., 2017; Wang et al., 2022).

In the Great Bay Area (GBA; including the Pearl River Delta, Hong Kong, and Macao), China, ozone pollution is closely linked to the western Pacific subtropical high (WPSH) and northwest Pacific typhoon in summer and autumn (Gong and Liao, 2019; Shao et al., 2022; Liu et al., 2023). In particular, when a typhoon approaches this region, severe ozone pollution easily occurs (Deng et al., 2019; Qu et al., 2021). Many previous studies have suggested that when the GBA is at the

periphery of a typhoon (Hu et al., 2010; Jiang et al., 2015), high solar irradiance and high temperature promote the ozone photochemical production (So and Wang, 2003; Ding et al., 2004), while unfavorable diffusion conditions increase the accumulation of ozone (Jiang et al., 2008; Li et al., 2022). Thus, high-ozone episodes may occur and persist in the period leading up to a typhoon landfall. Numerical simulations can help us quantitatively study the evolution of this type of ozone pollution (Li et al., 2022; Ouyang et al., 2022; Wang et al., 2022). Furthermore, numerical simulations can also help us

separate the impacts of typhoons by comparing base simulation with sensitive simulation that remove the typhoon system (Hu et al., 2019). However, two key issues must be considered, as they may directly affect the accuracy and credibility of the simulation results. The first problem (Prob. 1) is related to whether the model can capture the variations in this type of ozone pollution, especially the extremely high ozone concentration. The second problem (Prob. 2) occurs because this type of ozone pollution is controlled by the competitive effects of the WPSH and typhoon; hence, erasing the typhoon system may

increase the impact of the WPSH, which may lead to a high bias in the simulation results or even affect the final conclusions (Gilliam et al., 2015; Chatani and Sharma, 2018).

Ensemble simulation by perturbing meteorological factors in the initial and boundary conditions can offer a solution to the above problems. This method has a greater probability to capture the spatial and temporal variations of air pollutants in extreme synoptic systems (Delle Monache et al., 2006; Zhang et al., 2007; Bei et al., 2014; Zhu et al., 2016), thereby

providing more accurate data for relevant studies and yielding more reliable conclusions. More importantly, by comparing well-performing ensemble members (good EMs) with poorly performing EMs (bad EMs), we can better understand the impact of meteorology on ozone, which may help mitigate the uncertainties associated with Prob.2.

In this study, we applied the Weather Research and Forecasting model (WRF; Skamarock et al., 2008) with Chemistry (WRF-Chem; Grell et al., 2005) coupled with an ozone source apportionment method (WRF-Chem-O3tag; Gao et al., 2016;

2017). By perturbing meteorological factors in the initial and boundary condition files, we constructed an ensemble simulation (including 30 EMs) to simulate the extremely high-ozone episode that occurred in the GBA associated with the approach of Typhoon NIDA. By comparing the results of the EMs, we aimed to address the following issues: (1) more accurately capture the variation pattern and the extremely high concentration of surface ozone in the GBA; (2) elucidate the physical and chemical formation mechanisms of this ozone episode; and (3) quantify the changes in the ozone contributions

from various geographical source regions during this pollution episode. The structure of this paper is as follows: Section 2





provides a description of the ozone episode, model system and ensemble simulation. The results and discussion are presented in section 3. The conclusions are summarized in section 4.

## 2 The high ozone episode and the methods applied in this study

### 2.1 The extremely high-ozone episode and the basic simulation

#### 2.1.1 Severe ozone pollution when Typhoon NIDA approached

Between July 28 and August 3, 2016, Super Typhoon NIDA formed in the western Pacific and eventually made landfall along the coast of the GBA, resulting in severe disasters in South China, particularly in Guangdong and Guangxi Provinces. When Typhoon NIDA approached the GBA, a high-ozone episode occurred in this region. During the afternoon of July 31, the mean concentrations of surface ozone exceeded 200 μg m$^{-3}$ (grade II of the national standard for the hourly ozone

concentration) in most areas of the GBA (black square). Notably, the maximum concentration reached 366 μg m$^{-3}$, recorded in Jiangmen city.

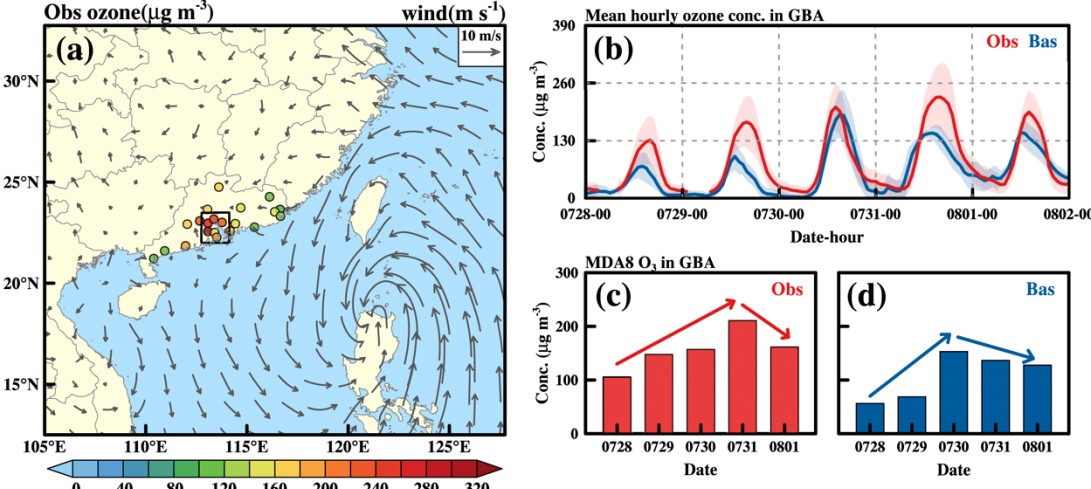

**Figure 1: (a) Ozone observations (Obs ozone) for Guangdong Province and the 10-m wind fields at 14:00 on July 31, 2016; (b)**
**mean hourly ozone concentrations in the Great Bay Area (GBA) from 00:00 on July 28 to 00:00 on August 2, 2016; (c-d) mean maximum daily 8-hour average ozone (MDA8 O₃) in the GBA from July 28 to August 1, 2016. The 10-m wind data originates from the European Centre for Medium-Range Weather Forecasts (ECMWF) Reanalysis V5 (ERA5) dataset. In Fig. 1(b), the shaded areas denote the standard deviations; red and blue denote the ozone observations (Obs) and basic simulation (Bas), respectively.**



### 2.1.2 Basic simulation by using WRF-Chem model

In this study, we conducted a basic simulation using the WRF-Chem model. This model is a fully coupled 3D Eulerian model system that incorporates two-way online feedback between the meteorological model and the chemical transport model. It has been widely employed in air quality research and forecasting studies (Li et al., 2018; Gao et al., 2021; Yang et al., 2022; Huang et al., 2020; 2023). Regarding the model configuration, we established nested model domains (Fig. 2a). To simulate the lifetimes of both the ozone episode and Typhoon NIDA, the parent domain (D01) covered most parts of East Asia and Southeast Asia. The simulation period spanned from 00:00 on July 25 to 00:00 on August 4, 2016 (UTC). The first two days were designed as the spin-up period. Other detailed model configurations and descriptions of relevant datasets can be found in the Supplementary Information.

As shown in Fig. 1b, the comparison of the hourly ozone concentrations revealed that the basic simulation results (Bas) did not show good agreement with the observations (Obs) in the GBA. Although Bas could capture the diurnal variation well, it failed to simulate the ozone peaks during this episode. In particular, Bas significantly underestimated the maximum concentration that occurred in the afternoon of July 31. Regarding the mean maximum daily 8-hour average ozone (MDA8 $O_3$), the maximum value was reached on July 31 in Obs (Fig. 1c), whereas in Bas, the maximum value was reached on 30 July (Fig. 1d). The results suggested that Bas could not reproduce this ozone episode well, although the model settings have been widely employed and shown to be effective in many other air quality modeling studies. Hence, by using the same model settings, we conducted an ensemble simulation in this study. We believe that this approach can enhance simulation accuracy and facilitate a better understanding of this high ozone episode.

### 2.2 Ensemble simulation and ozone source apportionment

For the ensemble simulation, all EMs were generated with the "cv3" background error covariance option in the WRF-3DVAR package via perturbation of the meteorological factors in the initial and boundary condition files. The horizontal wind components, potential temperature, and water vapor mixing ratio were perturbed with standard deviations of 2 m s-1, 1 K, and 0.5 g kg-1, respectively (Zhu et al., 2016; Xiao et al., 2023). This ensemble initialization method has been widely utilized in data assimilation and weather analysis (Meng and Zhang, 2008a, b; He et al., 2019; Chan et al., 2022). In this study, we used this method to create 30 EMs and run them separately. By comparing the simulated ozone concentrations with the observations, "good" and "bad" EMs could be distinguished on the basis of their performance. The formation of the ozone episode and its source contributions could then be quantitatively examined by contrasting the results of the good and bad EMs.



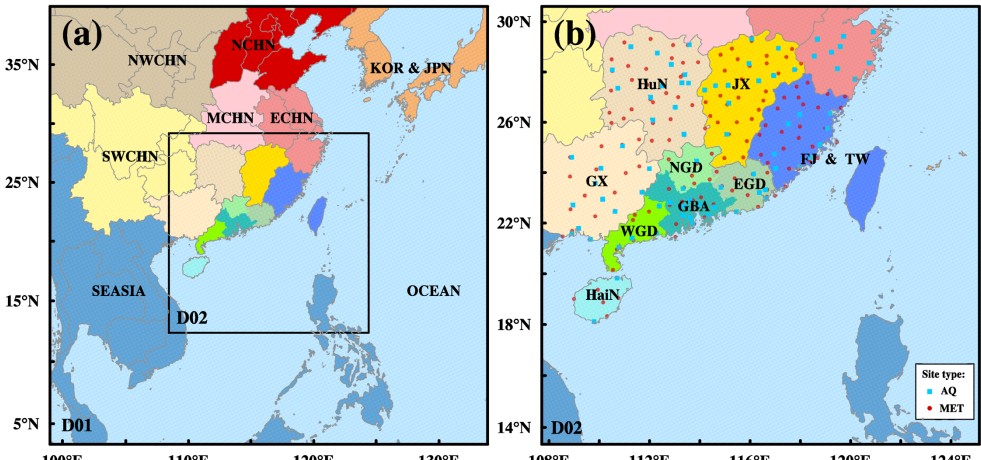

**Figure 2: Model domain and geographical source region setting. The locations of the air quality (AQ; square) and meteorology (MET; dot) observation stations are shown in (b).**

## 2.3 Online ozone source apportionment method coupled with the WRF-Chem model

In this study, we also employed our ozone source apportionment method in the ensemble simulation to quantify the ozone

contributions during this episode. Similar to the Ozone Source Apportionment Technology (OSAT; Yarwood et al., 1996), our method is a mass balance technique that aims to identify the ozone contributions from each geographical source region preset in the model domain. Due to its secondary pollutant properties, the photochemical production of ozone is attributed to the contributions of ozone precursors (NOx and VOCs) emitted from the geographical source regions. In this method, the model domain is divided into several source regions, where ozone and its precursors are treated as independent variables.

Because of the nonlinear relationship of ozone photochemistry, the ozone formation sensitivity (NOx-limited or VOCs-limited) will be determined for each grid at each time step. Then, the chemical production of ozone will be allocated based on the concentration ratio of the key ozone precursor (NOx for NOx-limited and VOCs for VOCs-limited) from each source region. These variables will also undergo all the physical processes as the original variables experience during the simulation, but they will not interfere with the original simulation. In addition, the initial and boundary chemical conditions of the

relevant species also need to be defined as sources to maintain the mass balance. More information on this method can be found in Gao et al. (2016, 2017).

Before initiating the simulation, we need to establish geographical source regions for the ozone apportionment method. There were 17 geographical source regions in the model domain (Fig. 2). The GBA, where we studied in this paper, was designated as one region. The other cities in Guangdong Province were categorized into three regions based on their relative

locations to the GBA: east Guangdong (EGD), north Guangdong (NGD) and west Guangdong (WGD). For other areas in China, the regional settings were defined according to administrative divisions. Source regions outside of China were





defined based on national divisions. The marine areas in the model domain were established as one region. The detailed compositions of the various geographical source regions are listed in Table 1. In addition, the boundary conditions were defined as one region. which we referred to as $O_3$ inflow, as it can flow into the model domain and have a significant impact on ozone concentrations, which is typically treated as background ozone (Gao et al., 2017; 2020). The initial conditions of D01 and D02 were also defined as independent ozone contributions (INIT1 and INIT2, respectively).

**Table 1: List of the geographical source regions in the model domain.**

| Source regions | Details |
| --- | --- |
| GBA | Pearl River Delta, Hong Kong and Macao, China |
| EGD | Cities in the east of Guangdong Province, China |
| NGD | Cities in the north of Guangdong Province, China |
| WGD | Cities in the west of Guangdong Province, China |
| FJ & TW | Fujian and Taiwan provinces, China |
| JX | Jiangxi Province, China |
| HuN | Hunan Province, China |
| GX | Guangxi Province, China |
| HaiN | Hainan Province, China |
| ECHN | East China, including Jiangsu, Zhejiang, Anhui Province and Shanghai city, China |
| MCHN | Middle China, including Henan and Hubei Province, China |
| SWCHN | Southwest China, including Yunnan, Sichuan, Guizhou, Xizang Province and Chongqing, China |
| NCHN | North China, including Hebei, Shanxi, Shandong, Liaoning, Jilin Province and Beijing, Tianjin cities |
| NWCHN | Northwest China, including Shaanxi, Inner Mongolia, Ningxia, Gansu, and Qinghai provinces |
| KOR & JPN | North Korea, South Korea, and Japan |
| SEASIA | Southeast Asia, including Brunei, Cambodia, India, Indonesia, Laos, Malaysia, Myanmar, Philippines, Thailand, and Vietnam |
| OCEAN | Ocean area, including the Bohai sea, Huanghai sea, Donghai sea, Nanhai, and parts of the western Pacific |



## 3. Results and Discussion

### 3.1 EM selection and model validation

By comparing the observed ozone concentration with those simulated by each EM, we identified the EMs that could well reproduce the high ozone episode in the GBA by considering the following two rules: (1) whether the EM could reproduce the variation pattern of the mean surface ozone in the GBA and (2) whether the EM could capture the maximum concentration that occurred in the afternoon of July 31. Thus, two statistical metrics were determined to select good and bad EMs: the correlation coefficient (R) of the mean surface ozone in the GBA between the simulation and observation and the mean normalized bias (MNB) calculated from the simulated and observed ozone concentrations during the afternoon of July 31 (12:00~16:00), when the maximum concentration occurred. In addition, using R and MNB intervals of [-1, 1] and [-100, 100], respectively, we introduced an index ($I_{dis}$) to quantitatively assess the performance of each EM. For each EM k, the index $I_{dis}^k$ can be expressed as:

$$I_{dis}^k = sqrt[(R^k \times 100 - R_{best} \times 100)^2 + (MNB^k - MNB_{best})^2], \tag{1}$$

where $R^k$ and $MNB^k$ are the correlation coefficient and MNB between the observed ozone concentration and the simulated ozone concentration obtained from EM $k$, respectively. The values of $R_{best}$ and $MNB_{best}$ are 1 and 0, respectively. Thus, as shown in Fig. 3a, $I_{dis}^k$, calculated using equation (1), represents the "distance" of the model performance of EM $k$ from the best performance (red star). A smaller $I_{dis}$ value indicates better performance of the EM. For clarity, $I_{dis}^k$ is also represented by colors in Fig. 3a, where red indicates that the index is close to the "best", while blue indicates that the index is far from the "best".

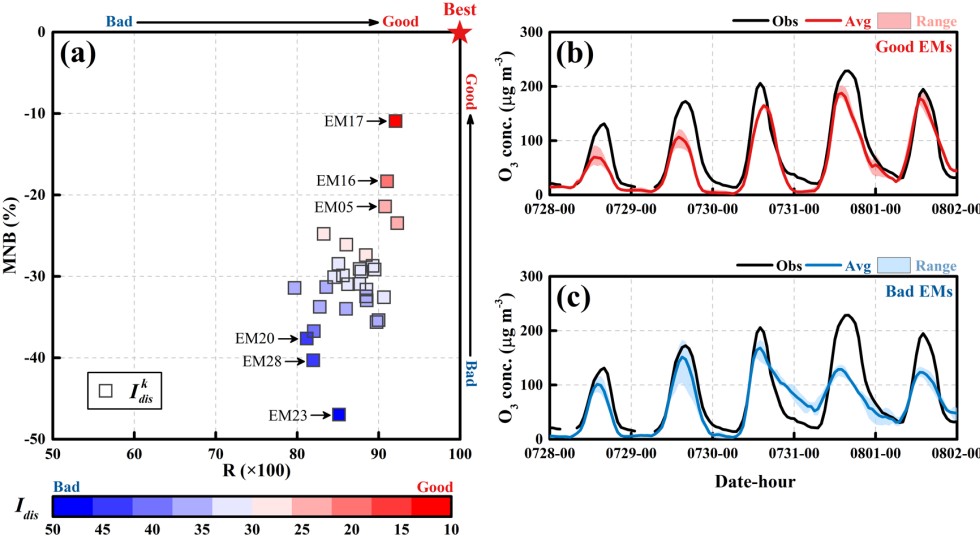





**Figure 3: (a) Model performance of each EM on surface ozone in the GBA; comparisons between the observed and simulated surface ozone concentrations in the GBA for (b) good EMs and (c) bad EMs.**

In Fig. 3a, $I_{dis}$ (square) indicated that EM05, EM16, and EM17 were the top three members with performances closest to the best, suggesting that these three EMs more effectively reproduce the variations in ozone in the GBA. In contrast, the $I_{dis}$ values of EM20, EM23, and EM28 were higher than those of the other EMs, indicating that they performed worse. Consequently, we classified EM05, EM16, and EM17 as good members, while EM20, EM23, and EM28 were classified as bad members. As shown in Fig. 3b, the comparison between the mean results of good EMs and the observations clearly revealed that the ozone time series of the good EMs agreed well with the observations and basically reproduced the variation pattern of ozone in the GBA during this episode, particularly the maximum concentration that occurred on July 31. For the bad EMs (Fig. 3c), time series of surface ozone exhibited an acceptable diurnal variation. However, they failed to capture the maximum ozone concentration on July 31. In addition, for both the good and bad EMs, we also did the model validations on meteorological factors (temperature at 2 m, T2; wind speed, WS; and wind direction, WD) and $NO_2$ from hundreds of stations located in middle and southern China (Table S3). The statistical metrics revealed that, both inside and outside the GBA, the good EMs demonstrated better model performance than the bad EMs. For example, all the meteorological factors of the good EMs exhibited higher indices of agreement (IOAs) than did those of the bad EMs. Most variables of the good EMs exhibited lower biases than those of the bad EMs in all regions, except T2 outside the GBA and WD in all regions. The good EMs exhibited a satisfactory model performance and could effectively reproduce the high-ozone episode in the GBA.

## 3.2 Physical and chemical causes of the extremely high ozone concentration on July 31

### 3.2.1 Process analysis of the average time series of surface ozone in the GBA

For surface ozone in the GBA, the average time series of the good and bad EMs (abbreviated as ozone_good and ozone_bad, respectively) are shown in Fig. 4a. The time series exhibited similar diurnal variations and concentrations from July 28 to the afternoon of July 30. The two ozone time series then began to vary differently, and the differences increased until the afternoon of July 31. Specifically, compared with that of ozone_bad, the time series of ozone_good declined more sharply in the afternoon of July 30, and the concentration remained relatively low until the early morning of July 31 (Stage I; light orange shaded area). From 08:00 on July 31, the time series of both ozone_good and ozone_bad began to increase, and the trends persisted until the afternoon, when high ozone concentrations occurred (Stage II; dark orange shaded area). Notably, ozone_good increased much more than did that of ozone_bad, resulting in the maximum concentration during this high-ozone episode in the GBA. The variation pattern was also consistent with that in the observations. As mentioned above, the relatively high maximum concentration of ozone_good may be attributed to the significant differences between ozone_good and ozone_bad from the afternoon of July 30 to the afternoon of July 31.



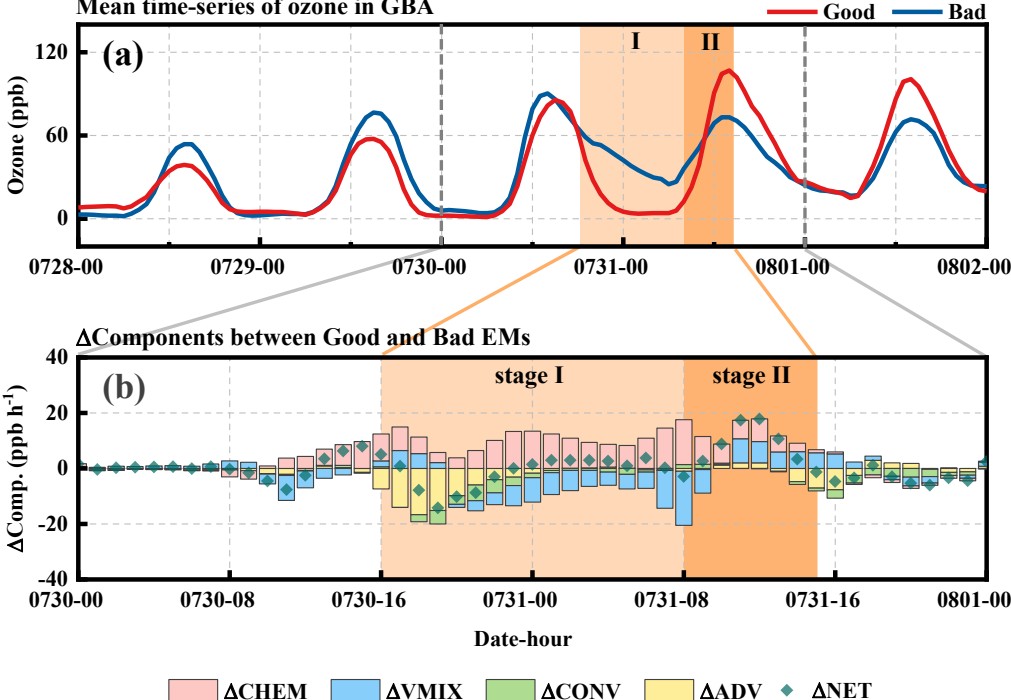

**Figure 4: (a)** Averaged time series of ozone_good and ozone_bad in the GBA and **(b)** the differences in each ozone component between the good and bad EMs from July 30 to 31. CHEM = chemistry, VMIX = vertical mixing, CONV = convection, ADV = advection, NET = net contribution.

The changes in ozone components (good-bad) could help us clarify the chemical and physical causes of the extremely high ozone concentration on July 31 (Fig. 4b). ΔNET is calculated as the sum of all differences from each process between the good and bad EMs, which can represent the relative changes in ozone_good relative to ozone_bad. For example, when both ozone_good and ozone_bad increased at Stage II, the positive ΔNET value indicated a more significant increase in ozone_good. At these two stages, ΔCHEM, ΔADV and ΔVMIX significantly contributed to ΔNET. At Stage I, although both ozone_good and ozone_bad decreased, the negative values of ΔNET during 18:00~21:00 on July 30 suggested that ozone_good decreased more significantly, which was attributed primarily to ΔADV. At Stage II, ozone_good increased more than ozone_bad did since ΔNET became positive. The budget of ΔNET suggested that the significant increase in ozone_good resulted from the combined effects of ΔCHEM and ΔVMIX. Hence, it could be concluded that the extremely high concentration of ozone_good that occurred in the afternoon of July 31 could be directly attributed to the ozone contributions from chemical and vertical mixing processes. In addition, ΔADV during the evening of the previous day also contributed to the extremely high concentration. In the following section, we will discuss the causes of the extremely high ozone concentration from physical and chemical perspectives.

...
...



## 3.2.2 Physical cause of the extremely high ozone concentration

Among the physical processes, VMIX was the direct contributor to the high concentration of ozone_good. By comparing the VMIX of ozone_good with that of ozone_bad (ΔVMIX in Fig. 5a1), the negative values aloft and positive values near the surface suggested that more ozone from ozone_good was entrained downward to the surface, leading to a greater increase in ozone at ground level. As the two key factors of VMIX (Gao et al., 2018), the differences in the vertical exchange coefficient and the vertical ozone profile between the good and bad EMs could explain the significant contribution of VMIX in good EMs. In the morning of July 31, there were no obvious differences between the good and bad EMs (Fig. 5a2). However, the vertical ozone profiles (Fig. 5a3) revealed that ozone_good exhibited higher vertical gradients from the surface to the top of the PBL. Under these conditions, compared with the bad EMs, the good EMs showed more transport of ozone from the top of PBL to surface, despite similar vertical exchange intensities.

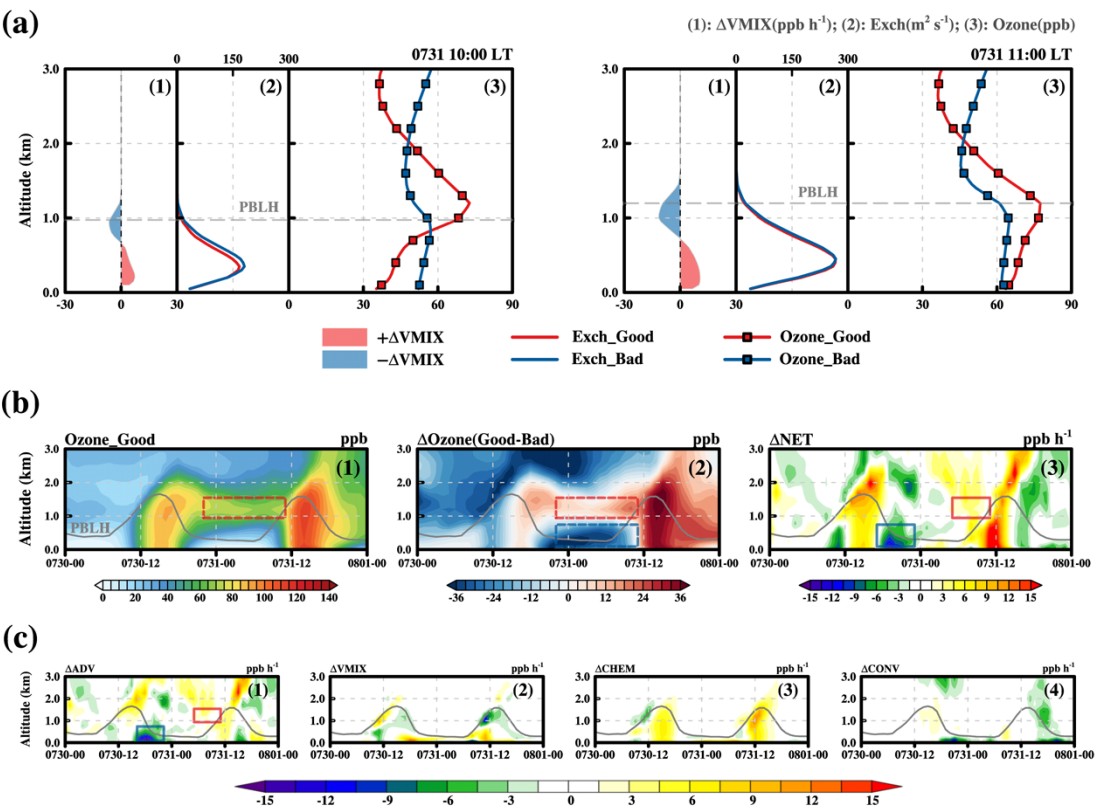

**Figure 5: (a) Profiles of (1) ΔVMIX, (2) exchange coefficient and (3) ozone at 10:00 and 11:00 on July 31; (b) vertical distributions of (1) ozone_good and the changes in ozone [(2) ΔOzone] and NET [(3) ΔNET]; (c) vertical distributions of the changes in the ozone contribution of each process [(1) ΔADV; (2) ΔVMIX; (3) ΔCHEM; (4) ΔCONV].**



The vertical distributions of ozone_good (Fig. 5b1) revealed that the vertical gradient persisted from the night of July 30 to the next morning, which is consistent with the vertical structure of ozone concentrations reported in other urban areas (Xu et al., 2018; Gao et al., 2020). The difference between ozone_good and ozone_bad (ΔOzone in Fig. 5b2) showed that the vertical gradient of ozone_good was even greater, as positive ΔOzone values occurred aloft and since negative values occurred in the lower layers during this period. Notably, the positive ΔOzone values aloft were even greater during the morning of July 31, which could increase the vertical gradient of ozone. ΔNET (Fig. 5b3) revealed a negative zone (blue square) at altitudes of 0–750 m from 18:00 to 23:00 on July 30 and a positive zone (red square) at altitudes of 1000–1500 m from 05:00 to 11:00 on July 31. These two changes were the direct causes of the vertical structure. By separating ΔNET into changes in each ozone component (Fig. 5c1–4), it could be observed that ΔADV exhibited the same features (the negative and positive zones in Fig. 5c1) as did ΔNET at the same altitude during the same period, suggesting that the significant vertical gradients of ozone_good were primarily caused by distinct ΔADV values at different levels during various periods.

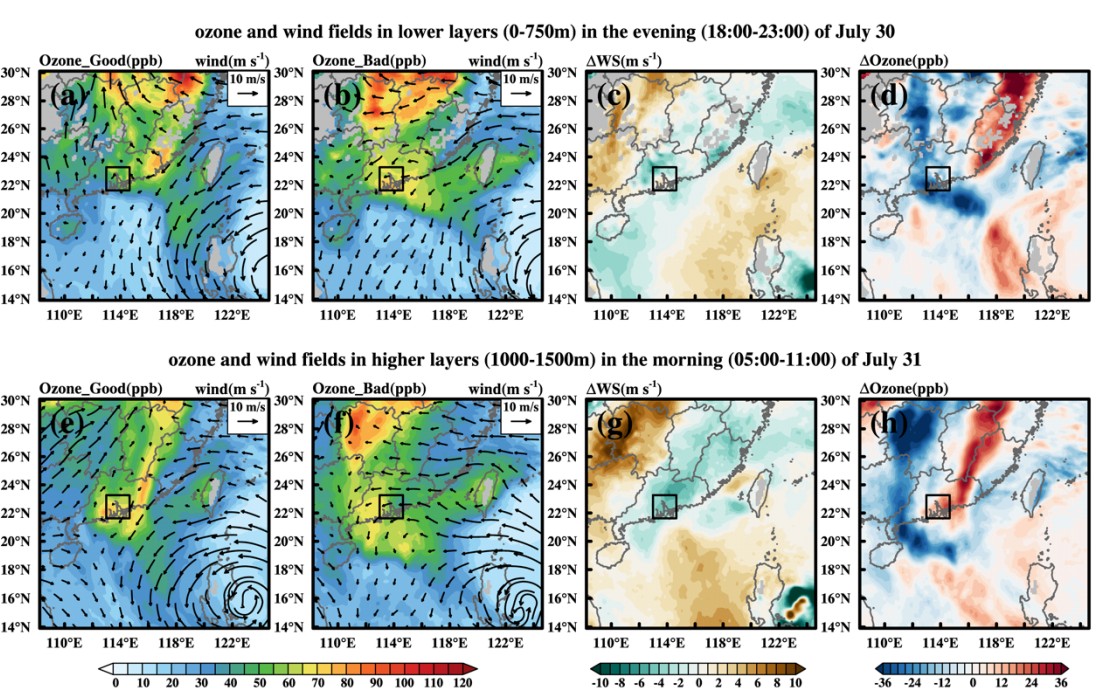

**Figure 6: Mean distributions of the ozone concentration and wind fields and their respective differences between the good and bad EMs in the lower (a-d) and higher layers (e-h).**

ADV is influenced wind fields and ozone distributions (Gao et al., 2017). The mean distributions of the ozone concentration and wind fields are shown in Fig. 6. During the evening of July 30, the good EMs indicated southwest winds in the GBA in the lower layers (Fig. 6a). With the transport of southwesterly winds, the low concentrations of ozone over sea areas (upwind regions) could be transported to the GBA, leading to a decrease in ozone. However, the bad EMs showed that the GBA was



controlled by easterly winds (Fig. 6b). The ozone in the upwind region was approximately equal to or even higher than that in the GBA, which did not cause a significant decrease in ozone levels. Thus, the concentrations of ozone_good were lower than those of ozone_bad in the lower layers of the GBA in the evening of July 30 (Fig. 6d). The significant vertical ozone gradient was more closely associated with the increase in ozone aloft in the morning of July 31 (Fig. 6e-6h). In the higher

255   layers, the good EMs exhibited high ozone concentrations and approximate static winds across the GBA, which was favorable for ozone retention (Fig. 6e). However, the bad EMs exhibited stronger east winds but lower ozone concentrations in upwind regions, which might have led to a decrease in ozone over the GBA (Fig. 6f). Thus, the concentrations of ozone_good were higher than those of ozone_bad over the GBA during the morning of July 31 (Fig. 6h). In addition, the good EMs showed weaker winds than the bad EMs did at the two stages (Fig. 6c and 6g, respectively). Especially in the

260   higher layers, the weak winds played an important role in the increased ΔOzone value. We collected the observations of vertical WS from four stations in the GBA (the station codes and locations are listed in Table S2 in the Supplementary Information). As shown in Fig. 7a, the observations revealed a low WS value at heights ranging from the surface to nearly 1700 m (dashed red square), which could confine ozone in the residual layer from the evening of July 30 to the early morning of July 31. Comparison of the simulations and observations revealed that the vertical wind of the good EMs agreed

265   better with the observations since the differences were relatively small in the lowest range of 2 km (Fig. 7b). In contrast, the bad EMs performed worse. Most of the time, the WS value of the bad EMs was greater than that of the observations, especially during the evening of July 30 and the morning of July 31 (Fig. 7c). By comparing the differences (Fig. 7b and 7c), especially in areas (solid squares) where ADV significantly differed (mentioned in Fig. 5c1), the WS value of the good EMs was closer to the observations, which not only indicates the better model performance of the good EMs for the vertical wind

270   features but also indicates the important contribution of static winds to the formation of high ozone levels in the residual layer.

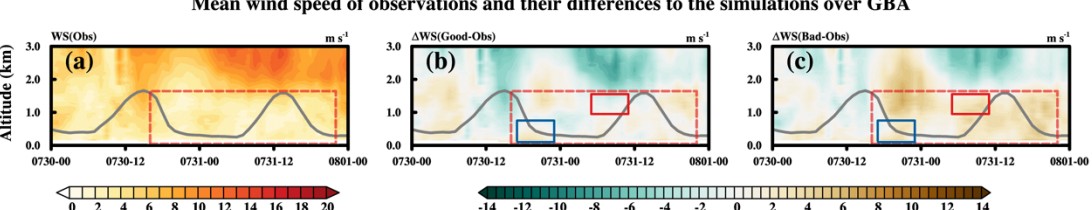

**Figure 7: Vertical distributions of the WS observations in the GBA (a) and differences of WS between the simulations and** 275   **observations [(b) for good EMs and (c) for bad EMs].**

### 3.2.3 Chemical cause of the extremely high ozone concentration

Enhanced CHEM was another key contributor to the extremely high ozone concentration. The changes in CHEM of ozone (ΔCHEM_Ozone) were positive within the PBL from 11:00 to 16:00 (Fig. 8a), suggesting that the photochemistry in the



good EMs was significantly greater than that in the bad EMs. Notably, enhanced photochemistry was more significant in the middle and upper layers of the PBL. Correspondingly, CHEM of NOx (ΔCHEM_NOx) in the good EMs decreased significantly, as indicated by the negative values ΔCHEM_NOx (Fig. 8b), which also indicated that enhanced photochemistry occurred in the good EMs.

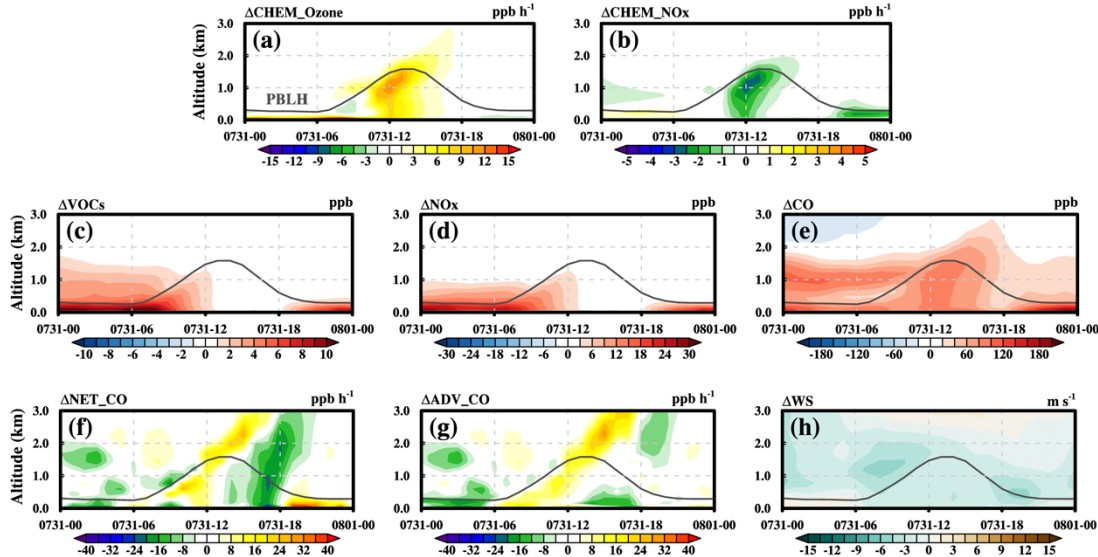

**Figure 8: Vertical distributions of the changes in CHEM of ozone and the induced factors on July 31. (a) Changes in CHEM of ozone; (b) changes in CHEM of NOx; (c) changes in the concentrations of VOCs; (d) changes in the concentrations of NOx; (e) changes in the concentrations of CO; (f) changes in NET of CO; (g) changes in ADV of CO; and (h) changes in the WS.**

The photochemical production of ozone primarily depends on ozone precursors and chemical reaction constants (Monks et al., 2015). Although the meteorological factors differed significantly between the good and bad EMs, the relevant photochemical reaction constants, such as the photolysis rates (Fig. S2), did not show significant differences in the PBL during daytime. Thus, the enhanced photochemical production of ozone in the good EMs could be attributed primarily to the increase in ozone precursors. As shown in Fig. 8c and 8d, the changes in NOx (ΔNOx) and VOCs (ΔVOCs) suggested that the good EMs contained more ozone precursors within the PBL in the morning of July 31, which could enhance ozone photochemistry. Interestingly, ΔNOx and ΔVOCs could approach 0 ppb in the afternoon, which did not align with the significant increase in ΔCHEM_Ozone during this period. This discrepancy arises because photochemical reactions are rapid, leading to the swift consumption of ozone precursors during their accumulation. Thus, although the meteorological conditions favored the accumulation of ozone precursors, they were consumed quickly by photochemical reactions.

To better characterize the accumulation of ozone precursors, we applied CO as a substitution. Since CO changes induced by chemistry are much lower than those induced by physical processes, the distributions of CO better reflect the accumulation of gaseous pollutants (Ding et al., 2013b). As shown in Fig. 8e, the concentrations of CO in the good EMs were much higher



than those in the bad EMs within the PBL not only in the morning but also in the afternoon. By comparing the changes in NET of CO (Fig. 8f) with those in ADV of CO (Fig. 8g), it could be concluded that the increased CO was primarily

attributed to the advection process within the PBL during daytime. The enhanced ADV of CO was also due to weak winds. As shown in Fig. 8h, the WS in the good EMs was lower than that in the bad EMs within the lowest 2 km during daytime. Lower WS favored the accumulation of gaseous pollutants, particularly ozone precursors, which could subsequently enhance the ozone photochemistry.

**Mean ΔT2 and ΔBE_ISOP in the morning of July 31**


**Figure 9: Mean distributions of ΔT2 (a) and ΔBE_ISOP (b) during the morning (09:00~12:00) of July 31.**

In addition, high temperature is a key factor in enhancing biogenic emissions, which can increase the concentrations of VOCs in the atmosphere. Comparative analysis showed that the temperatures in the good EMs were higher in most parts of

Guangdong (GD) and Jiangxi (JX) Provinces (Fig. 9a). Consequently, increased temperature could lead to the enlargement of leaf stomata, thereby emitting more biogenic VOCs (BVOCs) into the atmosphere (Guenther et al., 2006; 2012). Adopting isoprene as an example (Fig. 9b), the changes in the isoprene induced by biogenic emissions (ΔBE_ISOP) closely matched the changes in T2 (ΔT2), indicating that the higher temperature in the good EMs resulted in increased biogenic VOCs in the atmosphere, especially in the western and northern parts of GD. Northwest winds facilitated the transport of more VOCs to

the GBA, promoting the photochemical production of ozone, which is another important chemical cause of the extremely high ozone concentration on July 31. Our results support the findings of a previous study that highlighted the impact of BVOCs on ozone pollution in South China during typhoon approaches (Wang et al., 2022).




### 3.3 Source apportionment of extremely high ozone concentrations

With the implementation of the ozone source apportionment method, the contributions of ozone from various geographical source regions on July 31 were quantified. As illustrated in Fig. 10a, the local area made the greatest contribution to surface in the GBA during the daytime. Especially during the high ozone period (11:00~16:00), the mean local contribution could reach 65.4 ppb. In addition to the GBA, the surrounding areas (i.e., WGD) and the remote source region (i.e., SEASIA) showed obvious ozone contributions (with mean contribution of 4.0 ppb and 6.8 ppb, respectively) during this period. In

addition, INIT2 and INFLOW, which derived from the initial and boundary conditions of chemistry, also contributed to the surface ozone in the GBA.

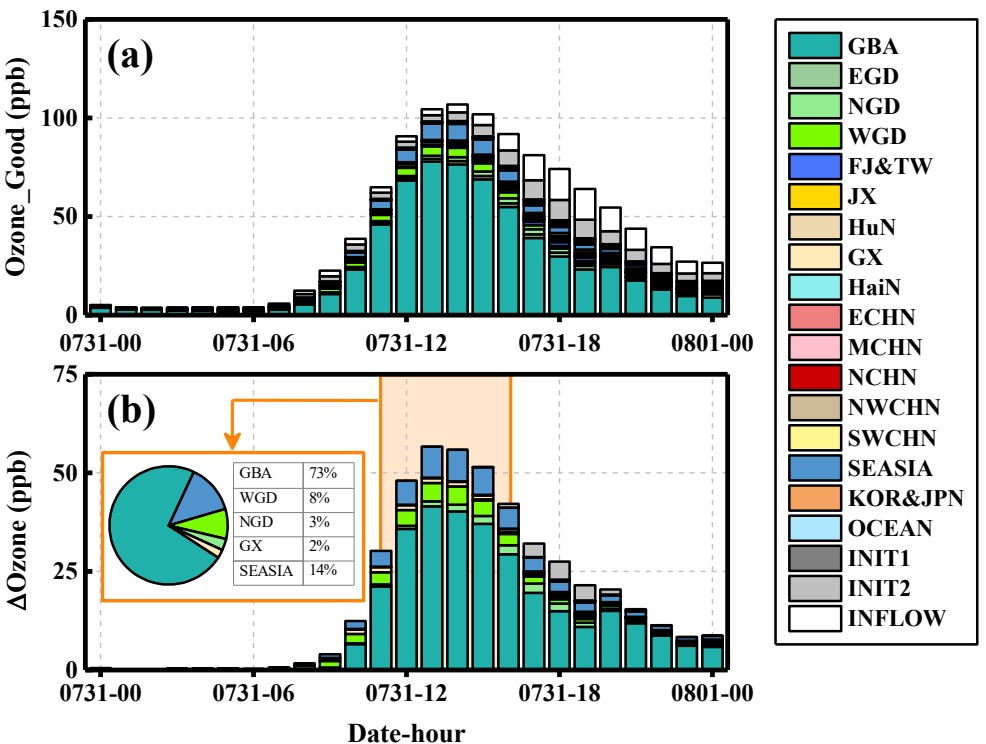

**Figure 10: Mean surface ozone contributions of good EMs (a) and mean changes in surface ozone contributions between the good**
**and bad EMs (b) in the GBA on July 31.**

By comparing with the bad EMs (Fig. 10b), the changes in ozone contributions were quantified, which could be used to determine the primary contributors to the increase in ozone during the daytime on July 31. It's clear that ozone contributions from the local region (GBA), surrounding regions (WGD, NGD and GX), and remote regions to the west (SEASIA)
increased on July 31. During the high-concentration period, the ozone contributions from these five source regions accounted for 73%, 8%, 3%, 2%, and 14%, respectively. In terms of local contribution, the GBA was controlled by stagnant weather,



which was not favorable for the diffusion of atmospheric pollutants. As a results, ozone precursors accumulated locally and produced more ozone through photochemical reactions. As shown in Fig. 9b, WGD, NGD and GX were controlled by westerly winds during the morning of July 31. Considering the greater amount of BVOCs emitted into the atmosphere over

these regions, more ozone and its precursors might be transported and contribute to the increase in surface ozone in the GBA. Our results are consistent with those of previous studies in this region (Li et al., 2012; 2022, Wang et al., 2022).

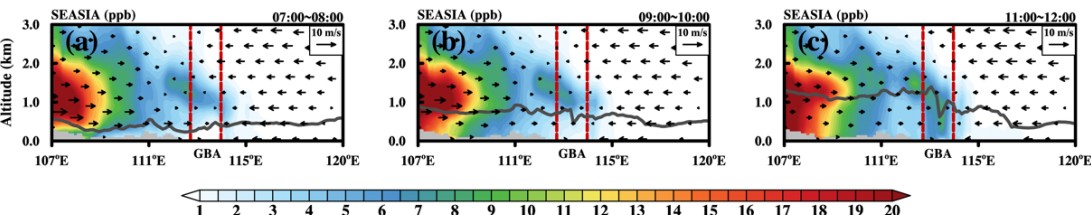

**Figure 11: Mean vertical cross sections of the ozone contribution from SEASIA along the west–east direction during the morning**
**on July 31. (a) 07:00~08:00; (b) 09:00~10:00; (c) 11:00~12:00.**

As a remote source region, the ozone contribution from SEASIA also significantly increased on July 31. However, unlike the local and surrounding regions, ozone and its precursors from SEASIA are challenging to transport to the GBA at ground level or low altitudes due to dry/wet deposition and chemical consumptions, suggesting a reliance on higher transport

pathways. Based on the vertical cross section of the ozone contribution from SEASIA along the west–east direction (Fig. 11), the high-concentration zone was located to the west of the GBA. Under the control of westerly winds, ozone from SEASIA could be transported to high altitudes in the area. Notably, due to the low wind speeds over the GBA, high ozone concentrations from SEASIA were confined to the residual layer in the early morning (Fig. 11a). When the PBL developed, the ozone aloft began to be entrained downward (Fig. 11b). With the continuous rise of the PBL, ozone aloft was kept on

being transported to surface through vertical mixing, ultimately contributing to increase surface ozone (Fig. 11c). This finding aligns with the physical cause of the high ozone concentration we discussed in section 3.2.2.

**4 Conclusion**

In the summer of 2016, an extremely high-ozone episode occurred in the GBA when Typhoon NIDA approached. The observations revealed that the maximum concentration reach as high as 366 µg m$^{-3}$ in the afternoon of July 31. Such

extremely high concentration poses a challenge for model simulations. To more accurately capture the pollution characteristics of this ozone episode, we conducted an ensemble simulation using the WRF-Chem-O3tag model. By using R and MNB, we introduced an index, $I_{dis}$, to select good and bad EMs based on their model performance on ozone. By comparing the good and bad EMs, we quantitatively examined the physical and chemical causes of the extremely high concentrations and the ozone contributions from various geographical source regions in this study.



The extremely high ozone concentration could be explained from both physical and chemical perspectives. From physical perspective, the GBA experienced weak winds as Typhoon NIDA approached. Weak winds facilitated a more significant accumulation of ozone in the residual layer during the evening and early morning. When the PBL developed in the morning, ozone aloft was continuously entrained downward to the surface and then significantly contributed to surface ozone. From chemical perspective, the approach of typhoon provided favorable meteorological conditions for ozone photochemistry, such

as increased temperature and more solar irradiance. However, our results revealed that the photochemical reaction constants (i.e., photolysis rate) changed only slightly during the high ozone period. More importantly, weak winds contributed to the accumulation of ozone precursors in the GBA. The increased ozone precursors were the primary reason for the dramatic enhancement of photochemistry. In addition, the increased temperature could increase the concentration of biogenic VOCs by intensifying biogenic emissions from vegetation. The increase in biogenic VOCs could also significantly contribute to the

increase in ozone precursors.

The ozone source apportionment results in the good EMs revealed that ozone from the local source region (GBA) contributed most significantly to the high ozone episode. Furthermore, the surrounding regions (i.e., WGD) and remote regions (i.e., SEASIA) also made significant contributions. During the high-concentration period on July 31, the mean ozone contributions from the GBA, WGD, and SEASIA were 65.4 ppb, 4.0 ppb, and 6.8 ppb, respectively. Compared to the bad

EMs, the ozone contributions from the GBA, WGD, NGD and GX significantly increased, which was primarily caused by stagnant weather in the GBA and westerly transport effects from outside the area. In addition, ozone from the western and remote region (SEASIA) could be transported to the GBA via westerly winds at high altitudes. When the PBL developed in the morning, ozone aloft was entrained downward, significantly contributing to surface ozone in the GBA.

**Data availability.** We have uploaded the data used in this paper to Zenodo. Please check it via the website https://doi.org/10.5281/zenodo.13868062

**Author contributions.** GJ and XH designed the research. XH prepared the input files for the ensemble simulation. GJ ran the model and analyzed the results. GJ wrote the paper. GJ and XH revised the paper.


**Competing interests.** The authors declare that they have no conflict of interest.

**Acknowledgement.** This work was supported by grants from the National Key Research and Development Program of China (2023YFC3709302); National Natural Science Foundation of China (42475104); Guangdong Basic and Applied Basic

Research Foundation (2023A1515011971); the Key Innovation Team of China Meteorological Administration (CMA2023ZD08); the Technological Innovation Capacity Enhancement Program of CUIT (KYQN202301); the Open Project Foundation of China Meteorological Administration Aerosol-Cloud and Precipitation Key Laboratory (KDW2402).



The authors acknowledge the Beijing Super Cloud Center (http://www.blsc.cn/) for providing HPC resources that have contributed to the numerical simulation in this study.

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
