# Peer review of "A Better Understanding of an Extremely High Ozone Episode with Ensemble Simulation"

_EGUsphere, 2024_

## Referee Comment (RC1)

The present article aims to investigate the physical and chemical drivers of extreme levels of Ozone in the Great Bay area of South China during the typhoon approach. The article shows the results of an intercomparison work of numerical simulations and the application of a source apportionment method to analyse the causes of extreme events and ozone high levels.

The content of the present manuscript is of interest to the scientific community both for the occurrence, more frequency of extreme weather events and their link with air pollution, and for the evaluation of the performance of CTM such as WRF-Chem in reproducing these events. From this point of view, the manuscript has certain relevance in the field and for the journal. On the other hand, I found the manuscript need more work to improve the clarity of the facts exposed and a better choice of images to show.

One of the starting questions reading the manuscript, is relative to the configuration of the model, the spatial resolution, and the inputs used for the weather and emissions. These elements should be mentioned in the text and in a table to let other scientists replicate the experiment in the future.

Secondly, the validation of the model performance should be more evidently quantitative. I appreciate the tentative to make the evaluation easy to understand by keywords (good/bad) and colours (red/blue) but it would be also good to know the real numbers behind the model performance. How is the best bias? How's the worst? These are numbers that in the most important intercomparison exercises are provided.

Said that I found interesting and intuitive the use of the Index linking MNB and R to show the model performance and I believe it's a kind of combined metric that would find replication in other works.

Finally, the images chosen for the manuscript are too small and too many. I suggest selecting those that need to stay in the main text and making them bigger in order to allow the reader to have a clearer way to examine them. Alternatively, I suggest combining the information provided in some multi-panel images in a small number.

**Major comments:**

- Section 3.2.1: How do the authors calculate the average time series that they show in Figure 4? Are they averaging the values in the whole GBA? Are they accounting only for the land part or also the water? Considering that the two models start to diverge in a night-day cycle could be possible to see also how the observations perform in that cycle?

- Section 3.2.2: Lines 250 - 255: this change in the wind direction between two simulations that should use the same re-analysis data as input of meteorology needs more clarification. Assuming the spin-up between the simulations is the same, the input data are the same and all the other parameters are the same, how do the authors justify this difference? Do the authors use any kind of nudging option in WRF to constrain the model outputs to the initial re-analysis fields?

- **Minor comments:**
  the authors should provide a more detailed description of the model configuration adopted for the simulations. In particular, there is information relative to spatial resolution, initial and boundary conditions used to feed the model, emission inventories for the chemistry and input of meteorology for WRF that should be mentioned and motivated in the methodology section to make the experiment replicable.

- Even if we appreciate that the authors are trying to make the validation easier to read it would be advisable to generate a range of confidence in model performance on the basis of the values of MNB, R and Index.

**Typos:**

Line 75: The authors mention " black square" but they don't mention Figure 1. Modify to (Figure 1, black square)

Line 133: Fig.2 to substitute into Fig.2 (b)

---

## Author Comment (AC1)

Dear referee #1:

Thank you very much for your thoughtful and insightful comments on this study. Your suggestions are very helpful to improve the quality of our manuscript. Based on your comments, we have revised the manuscript accordingly, including improving the clarity of the figures and providing the quantified results of the relevant statistical metrics. In addition, we have provided a sufficient introduction of model configuration and used datasets, which has been moved to supplementary information considering the length. Below, we provide a point-by-point response to your comments. The texts in blue are the comments, those in black are our response. The referenced line numbers correspond to the revised manuscript with changes marked.

**Major comments:**

1. Section 3.2.1: How do the authors calculate the average time series that they show in Figure 4? Are they averaging the values in the whole GBA? Are they accounting only for the land part or also the water? Considering that the two models start to diverge in a night-day cycle could be possible to see also how the observations perform in that cycle?

Reply: Thank you very much. During this ozone pollution in the GBA, high ozone concentrations were observed in the land region within the black square (Fig. R1a). Therefore, we selected simulated results from land-based grids to discuss the formation of the high ozone episode (Fig. R1b). In Fig. 4 of the original manuscript, the averaged time series were calculated using all simulated ozone concentrations from land grids within the black square. The water grids were not included in this study.

[Figure]

Figure R1. The observed ozone concentrations in Guangdong Province at 14:00 local time on July 31 (a); The selected grids used to discuss the formation of this high ozone episode in this study (b).

The mean time series of ozone_good and ozone_bad indeed began to diverge in the evening of July 30. Notably, ozone_good exhibited a more significant decrease during the nighttime and a stronger increase in the following morning compared to ozone_bad. To validate the simulations, we added the mean time series of the observed ozone (ozone_obs), calculated by averaging measured ozone from stations within the black square, into the Fig. R2. The ozone_obs similarly showed significant decrease in the evening of July 30 and a subsequent increase in the next morning. This pattern aligns more closely with ozone_good, indicating that the good EMs more accurately reproduced the diurnal cycle of ozone compared to the bad EMs.

[Figure]

Table R2. Mean time series of ozone in GBA from July 28 to August 02 in local time (LT). The observations, bad EMs and good EMs are denoted by grey, blue, and red lines, respectively.

2. Section 3.2.2: Lines 250 - 255: this change in the wind direction between two simulations that should use the same re-analysis data as input of meteorology needs more clarification. Assuming the spin-up between the simulations is the same, the input data are the same and all the other parameters are the same, how do the authors justify this difference? Do the authors use any kind of nudging option in WRF to constrain the model outputs to the initial re-analysis fields?

Reply: Thank you very much. In this study, we implemented the ensemble initialization method to conduct ensemble simulation (Meng and Zhang, 2008a, b). The initial and boundary condition files (wrfinput_d0* and wrfbdy_d01) from the base simulation were perturbed using the "cv3" background error covariance option in the WRF-3DVAR package. This procedure generated 30 new initial/boundary condition sets by randomly perturbing the meteorological variables: horizontal wind components, potential temperature, and water vapor mixing ratio. The perturbation magnitudes were constrained within one standard deviation of each variable (Zhu et al., 2016; Xiao et al., 2023).

These 30 perturbed condition sets were subsequently used to conduct the numerical simulation, producing 30 ensemble members (EMs). For the numerical simulation, we used identical reanalysis dataset and model configurations. In addition, nudging options were not applied. Divergences among EMs arose from the perturbations of initial/boundary conditions and the nonlinear error growth during numerical integration.

This ensemble simulation method has been utilized in data assimilation and weather analysis studies, and the findings have been widely validated and accepted by scientific community. In our study, we applied this method to improve the model performance ozone simulation during the extremely high ozone episode in GBA preceding the landfall of typhoon NIDA. Among the 30 EMs, several EMs performed better both in ozone concentrations and meteorological variables compared to the base simulation, which suggested that this method can effectively fulfill the demands of this study.

**Minor comments:**

1. The authors should provide a more detailed description of the model configuration

adopted for the simulations. In particular, there is information relative to spatial resolution, initial and boundary conditions used to feed the model, emission inventories for the chemistry and input of meteorology for WRF that should be mentioned and motivated in the methodology section to make the experiment replicable.

Reply: Thank you for your comment. We completely agree with you. In a modeling study, we believe that introducing the model configuration is crucial. We also welcome other scientists to replicate the experiment using our simulation method. We have provided an introduction to the model configuration for this study, which includes the domain setting, spatial resolution, parameter selection, and input datasets. Considering the length of the introduction, we have moved it to the supplementary information. Please refer to Text S1, Table S1, and Table S2 in the supplementary information for the relevant details.

2. Even if we appreciate that the authors are trying to make the validation easier to read it would be advisable to generate a range of confidence in model performance on the basis of the values of MNB, R and Index.

Reply: Thanks for this comment. The values of R, MNB and the Index of each EM are listed in Table R1. The R values range from 0.79 to 0.92, which suggested that all EMs could well reproduce the variation patterns of surface ozone in the GBA. However, the MNB values range from -46.99% to -10.92%. According to the recommended threshold values of MNB (±15%) provided by EPA (2005, 2007), only EM17 could meet the criterion, while the other EMs did not.

There is significant discrepancy in the validation result between R and MNB. To draw a common conclusion of the model performance in ozone, we introduced an index that quantitatively evaluates the performance of each EM. This index considered the effects of both R and MNB, which could evaluate the variation pattern and the high concentrations of ozone simultaneously. Based on the index, the top three EMs were EM05, EM16, and EM17, while the bottom three were EM20, EM23, and EM28.

Table R1. The R, MNB and Index of each EM on ozone in the GBA

| EMs | R (×100) | MNB (%) | $I_{dis}$ |
|---|---|---|---|
| EM01 | 86.23 | -30.97 | 33.90 |

| | | | |
|------|-------|--------|-------|
| EM02 | 88.44 | -27.40 | 29.74 |
| EM03 | 90.67 | -32.58 | 33.89 |
| EM04 | 83.25 | -24.80 | 29.93 |
| EM05 | 90.79 | -21.42 | 23.32 |
| EM06 | 89.75 | -35.62 | 37.07 |
| EM07 | 86.60 | -34.00 | 36.77 |
| EM08 | 82.77 | -33.74 | 37.88 |
| EM09 | 88.52 | -31.62 | 33.64 |
| EM10 | 85.64 | -29.89 | 33.16 |
| EM11 | 88.48 | -32.46 | 34.44 |
| EM12 | 82.06 | -36.74 | 40.89 |
| EM13 | 92.30 | -23.47 | 24.70 |
| EM14 | 90.01 | -35.37 | 36.76 |
| EM15 | 85.06 | -28.46 | 32.14 |
| EM16 | 91.03 | -18.35 | 20.43 |
| EM17 | 92.08 | -10.92 | 13.49 |
| EM18 | 87.63 | -30.95 | 33.33 |
| EM19 | 83.59 | -31.34 | 35.37 |
| EM20 | 81.18 | -37.64 | 42.08 |
| EM21 | 84.54 | -30.11 | 33.85 |
| EM22 | 89.51 | -29.16 | 30.99 |
| EM23 | 85.15 | -46.99 | 49.28 |
| EM24 | 86.04 | -26.11 | 29.60 |
| EM25 | 88.55 | -32.95 | 34.89 |
| EM26 | 87.68 | -29.10 | 31.60 |

| | | | |
|---|---|---|---|
| EM27 | 87.89 | -29.39 | 31.79 |
| EM28 | 81.99 | -40.32 | 44.16 |
| EM29 | 79.68 | -31.42 | 37.41 |
| EM30 | 89.28 | -28.72 | 30.66 |

Table R1 has been moved to the supplementary information, please check it there. In addition, the R and MNB values of each EM are also listed in the revised manuscript (Table 2). We have also added the relevant discussion on R and MNB in the revised manuscript. Please check the details at the lines 164-175.

**Typos:**

1. Line 75: The authors mention " black square" but they don't mention Figure 1. Modify to (Figure 1, black square)

Reply: Thank you very much. The figure 1 has been modified in accordance with the comment from referee #3. The black square is now in Fig. 1b. Thus, we have modified the "black square" to "black square in Fig .1b". Please check the details at line 75 in the revised manuscript.

2. Line 133: Fig.2 to substitute into Fig.2 (b)

Reply: Thank you for this comment. In the original manuscript, figure 2b did not cover all 17 geophysical source regions, whereas figure 2a did. In addition, figure 1 in the revised manuscript now presents the information about the source regions. Therefore, the "Fig. 2" has been modified to "Fig. 1a". Please check the details at line 144 in the revised manuscript.

**References**

Meng, Z. and Zhang, F.: Tests of an Ensemble Kalman Filter for Mesoscale and Regional-Scale Data Assimilation. Part IV: Comparison with 3DVAR in a Month-Long Experiment, Mon Weather Rev, 136, 3671-3682, 10.1175/2008mwr2270.1, 2008a.

Meng, Z. and Zhang, F.: Tests of an ensemble Kalman filter for mesoscale and regional-scale data assimilation. Part III: Comparison with 3DVAR in a real-data case study, Mon Weather Rev, 136, 522-540, 10.1175/2007mwr2106.1, 2008b.

Xiao, H., Liu, X. T., Li, H., Yue, Q., Feng, L., and Qu, J.: Extent of aerosol effect on the precipitation of squall lines: A case study in South China, Atmos Res, 292, 106886, 10.1016/j.atmosres.2023.106886, 2023.

Zhu, L., Wan, Q., Shen, X., Meng, Z., Zhang, F., Weng, Y., Sippel, J., Gao, Y., Zhang, Y., and Yue, J.: Prediction and Predictability of High-Impact Western Pacific Landfalling Tropical Cyclone Vicente (2012) through Convection-Permitting Ensemble Assimilation of Doppler Radar Velocity, Mon Weather Rev, 144, 21-43, 10.1175/Mwr-D-14-00403.1, 2016.

---

## Author Comment (AC2)

Dear referee #2:

Thank you very much for your comments on this work. The comments and suggestions are very useful to improve our manuscript. Here we provide a point-by-point response to your comments. The texts in blue are the comments, those in black are our response. The referenced line numbers correspond to the revised manuscript with changes marked.

**Specific Comments:**

1. While the index $I_{dis}^{k}$ (Eq. 1) effectively quantities EM performance, the threshold for classifying "good" and "bad" EMs is not explicitly defined. In addition, how does the index consider the temporal and spatial performance variations? For instance, some EMs might perform well during specific periods or under particular conditions but poorly in others.

Reply: Thank you very much. In this study, the index $I_{dis}^{k}$ represents the "distance" in model performance of an EM compared to the best performance. The smaller index value indicates better performance of the EM. therefore, we classified EMs with index ranking in the top 10% as "good" EMs, while those in the bottom 10% were classified as "bad" EMs. In this case, out of the 30 EMs, 3 EMs will be designated as good EMs and 3 as bad EMs. We believe this index can consider both temporal and spatial performance for the following reasons:

(1) For this ozone episode, the variation pattern of the ozone time series throughout the episode, along with the extremely high concentration observed in the afternoon of July 31, highlighted the temporal feature of this high ozone episode. These two factors are critical for evaluating whether the simulated ozone performed well. As a result, we implemented R to validate the ozone variation pattern and MNB to validate the high concentration in the afternoon of July 31. Since the index is calculated based on the contributions of both R and MNB, it incorporated the temporal features of the ozone episode.

(2) The ozone concentrations used to calculate R and MNB are derived from the average observed ozone across all stations within the GBA, as well as the average simulated ozone at corresponding grids. The average time series of ozone incorporated the spatial information of surface ozone in the GBA. Therefore, the index also reflects the spatial features of this episode to some extent.

2. More explanation and discussion on the construction of the $I_{dis}^k$ index is needed. Why were the correlation coefficient (R) and mean normalized bias (MNB) chosen? Although R and MNB were normalized for comparability, is it reasonable to assign equal weight to both R and MNB since the model can better capture the variation trend of ozone in most EMs?

Reply: Thank you very much. We believe that capturing the variation pattern of the ozone time series is one of the most important factors in demonstrating that the performance of an ozone simulation is "satisfactory" in a specific region. Both the correlation coefficient (R) and the index of agreement (IOA) can evaluate this feature, and since their effects are quite similar, choosing either is acceptable. In this study, we applied R.

For this ozone episode, the extremely high concentration of ozone that occurred in the afternoon of July 31 is another significant feature which the simulated ozone should well reproduce; however, the base simulation failed to capture this aspect. Many statistical metrics can evaluate this feather, for example mean bias (MB), root mean square error (RMSE) and mean normalized bias (MNB). We chose MNB for two reasons:

(1) MNB is a dimensionless quantity and has been normalized, making it similar to R.

(2) The EPA (2005, 2007) provides benchmarks of MNB (±15%), offering a clear reference for evaluating the performance of ozone simulation.

Since both the variation pattern and the extremely high concentration are key features of this high ozone episode. It is hard to determine which is more important. In this context, we assigned equal weight to both R and MNB to calculate the index $I_{dis}^k$. And according to the classification results derived from this index, comparing to the bad EMs and the base simulation, the good EMs demonstrated better model performance in this ozone episode, both in terms of the variation pattern and the high concentrations. This finding suggests that the index and the classification method are effective.

3. Here, the ensemble simulations were conducted with the perturbated meteorological fields. Hence, the physical and chemical discrepancy between the "good" and "bad" EMs is mainly attributed to the change in the meteorological fields. Emissions are another important factor affecting ozone formation. Although biogenic emissions were indirectly perturbed due to the change in meteorology, most emissions remain consistent in this study. This point should be pointed out and further discussed.

Reply: Thanks for this comment. We completely agree with your opinion. Among all the EMs, the only differences in input data and parameterization were the perturbed meteorological variables in the initial and boundary condition files. Therefore, the discrepancy between good and bad EMs can primarily be attributed to the changes in meteorological variables. Based on the model validations and the findings in our study, it is also suggested that more accurate simulations of meteorological variables lead to improved accuracy in the distribution and variations of ozone concentrations. In addition, it also should be noted that emissions are another important factor affecting ozone precursors, which can further affect the ozone through photochemistry. While emission indexes have improved substantially in recent years, there are still considerable uncertainties in them which may lead to uncertainty in ozone simulations. Quantifying the relationship between emissions and ozone concentration through ensemble simulation is also an interesting topic that we plan to study in our future work. Thank you once again for highlighting this point. We have added relevant discussions to the revised manuscript. Please check the details at lines 121-123 in the revised manuscript.

4. The number of ensemble members in this study is 30. How does the ensemble number was decided?

Reply: Thank you very much. Based on the work of Zhang et al. (2009), an ensemble size of 30 has been found to be affordable and reasonable for simulating typical cyclones. Therefore, we decided to conduct our ensemble simulation with 30 members in this study.

5. The index of the base simulation can be added into Figure 3. It could facilitate comparison between base and ensemble simulation.

Reply: Thank you very much. The index of the base simulation has been added into the new figure. Please check Figure 3 in the revised manuscript.

6. The caption of some figures needs to be modified for better understanding. For instance, "ΔCHEM" in Figure 5c, does it represent the differences between the CHEM of "good" and "bad" EMs? or just the contribution of the chemical process in the "good" EMs? Similar problem should be checked for other figures.

Reply: Thank you very much. We have double-checked all the figures relating to the issue you mentioned. To enhance clarity, we have added "Good-Bad" to the figures that illustrate the differences between good and bad EMs or their captions. Please check the

updated figures and captions in the revised manuscript.

7. Some of the subfigures are small and need to be adjusted, as in Figure 5c. It should be mentioned at least once that the time zone on the x-axis is local time.

Reply: Thank you for your comment. We have double-checked all the figures related to this issue. For the relevant figures, we have added "LT" (abbreviation for local time) to indicate that the time is in local time. Please check the updated figures in the revised manuscript.

**References**

EPA, U.S.: Guidance on the Use of Models and Other Analyses in Attainment Demonstrations for the 8-hour Ozone NAAQS, EPA-454/R-05-002, 2005.

EPA, U.S.: Guidance on the Use of Models and Other Analyses for Demonstrating Attainment of Air Quality Goals for Ozone, PM2.5, and Regional Haze, EPA-454/B-07-002, 2007.

Zhang, F., Weng, Y., Sippel, J.A., Meng, Z., and Bishop, C.: Cloud-resolving hurricane initialization and prediction through assimilation of Doppler radar observations with an ensemble Kalman filter, Monthly Weather Review, 137: 2105-2125, 10.1175/2009MWR2645.1 2009.

---

## Author Comment (AC3)

Dear CC#1:

Thank you very much for your comments on this work. Your feedback and suggestions are invaluable for improving our manuscript. Below, we provide a point-by-point response to your comments. The texts in blue represent the comments, while those in black are our responses. The referenced line numbers correspond to the revised manuscript with changes marked.

**Specific comments:**

1. The number of small figures in the manuscript is excessive. It is recommended to simplify the figures wherever possible:

(1) Figure 1a could be combined with Figure 2a. The wind field in Figure 1a should be moved to Figure 2a, while the O3 concentrations from the stations in Figure 1a could be placed in Figure 2b. In Figure 2b, the meteorological stations should be shown in black or gray. Additionally, consider swapping the positions of Figures 1 and 2.

Reply: Thank you for your comment. Based on your suggestions, we have made modifications to Figures 1 and 2. And the positions of the two figures were swapped as well. Please check the updated figures in the revised manuscript.

(2) The information in Figure 4a overlaps with that in Figures 3b and 3c. It is suggested to combine Figures 3b and 3c into one and present them as Figure 4a.

Reply: Thank you very much. The time series of ozone in Figures 3 and 4 represented different situations. Figures 3b and 3c emphasize comparisons between the observed ozone and the simulated ozone from good and bad EMs, respectively. For these figures, the ozone time series of observations was calculated by averaging the measured ozone across the stations in the GBA. Correspondingly, for the simulations (both good and bad EMs), we selected the simulated ozone from the grids nearest to the stations in the GBA and calculated the time series. The comparisons confirmed that the good EMs performed better in ozone simulation than the bad EMs did.

In Figure 4a, we aim to highlight the different variations of ozone between good EMs and bad EMs in the GBA. Therefore, the time series of ozone in Fig. 4a were calculated by averaging the ozone concentrations from all the land grids in the GBA. Through the comparisons, the differences in ozone variations between good EMs and bad EMs can be figured out clearly.

Thus, we believe it is appropriate to keep figure 3b, 3c and 4a as they are.

(3) There are too many subplots in Figure 5, which makes it difficult to interpret the figure references in the text. For example, it is unclear whether Fig. 5a1 corresponds to 10:00 LT or 11:00 LT. I recommend reducing the number of subplots.

Reply: Thank you very much. We have modified Figure 5. Firstly, the vertical profiles in Fig. 5a showed similar features at 10:00 and 11:00. We have used the profiles at 11:00 as an example to illustrate the differences in VMIX, EXCH and ozone concentrations between good and bad EMs, which are now presented as Figure 5 in the revised manuscript. Secondly, the other subfigures from the original Figure 5 have been simplified and are now presented as a new figure to explain why good EMs exhibit enhanced VMIX. The new figure is titled as "Figure 6". Please check the updated figures in the revised manuscript.

2. In Figure 3a, it would be better to use abs(MNB) on the Y-axis, as MNB can take both positive and negative values.

Reply: Thank you for your comment. The Y-axis of Figure 3a has been changed from "MNB" to "abs(MNB)". Please check the updated Figure 3a in the revised manuscript.

3. Line 247:"ADV is influenced wind field..." should be "The contribution of ADV is influenced by wind field …"

Reply: Thank you very much. The description has been modified. Please check the details at line 290 in the revised manuscript

4. Line 305: "The enhanced ADV of CO was also due to weak winds." This sentence may cause confusion. Strong winds accompanied by large positive/negative concentration gradients can lead to high positive/negative ADV contributions, while weak wind speeds result in lowerCO inflow or outflow. In Figure 8g, the positive ΔADV_CO could indicate that under good EMs, weaker winds have reduced the outflow of CO from the GBA region, which may be an important source of CO.

Reply: Thank you very much. We agree that discussing the change in CO contribution of ADV (ADV_CO) should consider both CO distributions and wind fields. In this study, good EMs showed enhanced ADV_CO during the daytime of July 31. We think it was primarily due to the weak winds. As shown in Fig. R1a, anthropogenic emissions of CO were high in the GBA (black square) and low in the surrounding areas. Consequently, when CO is emitted into the atmosphere, GBA is more likely to form a

high-concentration zone compared to the surrounding areas. Then, with the effects of wind fields, the high-concentration zone of CO will move towards the downwind area. In this condition, wind speed is key factor affecting CO concentrations in the GBA. Comparing the good EMs (Fig. R1b) to the bad EMs (Fig. R1c), the wind speeds of good EMs were lower than those of bad EMs. The lower wind speed could lead to the high concentration of CO remaining in the GBA for a longer duration. Therefore, it could be concluded that the enhanced ADV_CO of good EMs was primarily due to the weak winds

[Figure]

Figure R1. The mean distributions of the anthropogenic emission of CO (a) and the CO concentrations from surface to the height of 1500m during the afternoon (12:00~16:00 local time) of July 31. (b) for good EMs; (c) for bad EMs

5. Lines 338-341: "It's clear that … 14%, respectively." These sentences are unclear. Do the authors intend to convey that compared to the bad EMs, the good EMs show a significant increase in contributions from regions like GBA? Among the increased contributions, the proportions from various source regions are 73% … 14%?

Reply: Thank you very much. We apologize for this misunderstanding. What we aim to convey is that among all the source regions, the contributions from the five regions showed a significant increase. During the period from 11:00 to 16:00 LT, their respective proportions in the ozone increase account for 73%, 8%, 3%, 2%, and 14%. We have modified the relevant content, please check the details at lines 406-408 in the revised manuscript.

6. Many studies suggest that subsiding airflows in the periphery of typhoons can transport O3 from the upper troposphere or even the lower stratosphere to the surface. Did your simulations identify any similar vertical transport processes? If so, what is the contribution of this vertical transport to surface O3 levels?

Reply: Thank you very much. Previous studies have reported that the weather conditions are mostly sunny with lower wind speeds under the influence of peripheral subsiding airflows from typhoons. Such meteorological conditions can enhance the photochemical production of ozone, which is considered an important factor in the formation of high ozone episodes. In addition, as you mentioned, the subsiding airflows may also bring ozone from the upper troposphere or even the lower stratosphere down to the surface, contributing to ozone pollution. However, since the WRF-Chem model is not coupled with any stratospheric chemistry mechanisms, ozone at such altitude primarily comes from the chemical initial and boundary conditions. In this study, the two initial conditions (for domains 1 and 2, respectively) and the boundary conditions were set as independent source contributions (INIT1, INIT2, and INFLOW, respectively), but we did not further subdivide these contributions based on altitude. Therefore, ozone contribution from high altitudes (equal or higher than upper troposphere) cannot be quantified in this study. In addition, it is indeed an interesting topic. And if the WRF-Chem model can be coupled with stratospheric chemistry mechanisms, the quantitative results will be more accurate and meaningful. We think we may try this work in our future research.

---

## Author Response (AR2)

Dear Editor,

Thank you for your efficient handling of our manuscript. We sincerely appreciate your constructive suggestions, which can improve the quality and clarity of our manuscript. Below, we provide our responses to your suggestions. The texts in blue are the suggestions, those in black are our responses.

1. The title needs to be modified based on the ACP guidelines.

Reply: Thanks for your reminder. We are also very grateful for the title you suggested. We have revised the title in accordance with your recommendation, please check the details in the revised manuscript.

2. The conclusion needs to be modified based on the ACP guidelines.

Reply: Thank you very much for your feedback. We have revised the conclusion based on your comments. Please check the updated conclusion in the revised manuscript.